# Effectiveness of Climate-Smart Agriculture Innovations in Smallholder Agriculture System in Ethiopia

**Abyiot Teklu** [1,*], **Belay Simane** [1] and **Mintewab Bezabih** [2]

1   College of Development Studies, Addis Ababa University, Addis Ababa 1176, Ethiopia
2   Ethiopian Policy Studies Institute, Addis Ababa 2479, Ethiopia
*   Correspondence: abyiot.teklu@aau.edu.et

**Abstract:** The scientific basis for conceptualizing how farm households achieve the three climate-smart agriculture (CSA) pillars, the "triple benefit", is not well developed. This paper examined the impacts of CSA innovations on simultaneously enhancing food security, climate adaptation, and reducing GHG emissions. A cross-sectional household survey was collected from a multi-stage sample of 424 smallholder farmers selected from five agroecosystems of the upper Blue Nile highlands in Ethiopia and analyzed using an endogenous switching regression (ESR) model. CSA innovations, improved variety, compost, row planting, and agroforestry, provide farmers with the benefits of enhanced food security and climate change adaptation, reducing GHG emissions from farm plots. Crop rotation provides farmers with enhanced food security and reduced livelihood vulnerability, while SWC meets the goal of enhancing food security and reducing GHG emissions. Unfortunately, adopting crop residue management, one of the recommended CSA practices in Ethiopia, does not deliver at least two of the CSA pillars. Farmers should be encouraged to adopt improved variety, crop rotation, compost, row planting, soil and water conservation, and agroforestry as the best portfolio of CSA innovation for highland smallholder agriculture systems.

**Keywords:** climate-smart agriculture; livelihood vulnerability; food security; GHG emissions; synergy; trade-off; Ethiopia



## 1. Introduction

Current agricultural development policies, strategies, and technologies are not in sync with the global effort to mitigate current and future climate change trends, nor do they provide a window of opportunity for smallholder farmers to build climate resilience [1]. Climate change and agriculture are intertwined in three critical links. First, climate change affects crop yield and productivity, which causes food insecurity for smallholder farmers [2]. Second, for sustainable future agricultural development, agriculture should adapt to climate change to reduce climate-related livelihood vulnerability [3]. Third, agriculture is a source of greenhouse gas (GHG) emissions that accounts for thirteen, forty-four, and eighty-two percent of carbon dioxide, methane, and nitrous oxide emissions [4]. Therefore, future and current development policies should address the complex challenges of sustainable farming, food security, and the current and anticipated climate change [5,6].

Land degradation, low-input use, water scarcity, and a lack of adaptive capacity contribute to a decline in land productivity in Ethiopian smallholder agriculture [7–9]. When these issues are combined with the risk of climate change, the smallholder agriculture system suffers from additional productivity loss, increased water scarcity, and a lack of adaptive capacity to respond to climate change risk [10,11]. Whenever yield declines, smallholder households face malnutrition and food insecurity. Crops stored for the lean season may also be harmed due to crop pests caused by climate change. Furthermore, road infrastructure can be washed away or closed, restricting market access and aggravating food insecurity [12,13].

The shift of agricultural areas from more suitable land to marginal and environmentally fragile areas may exacerbate the problem of GHG emissions in three ways. First, degraded and abundant farmland emits more greenhouse gases than cultivated land. Second, the cultivation of grazing and wet land may release already stored carbon in the area and exacerbate GHG emissions [14]. Finally, when smallholders deforest natural forest area for cultivation, the GHG emissions released into the atmosphere increase.

Climate-smart agriculture (CSA) and livelihood diversification can improve the resilience and sustainability of African food systems in the face of climate change [15]. CSA integrates sustainable productivity, resilience (adaptation), emissions reductions, food security, and development objectives [6,16,17]. CSA innovations involve new and old agricultural technologies and are part of conservation agriculture [18–20]; agroforestry [21–26]; sustainable intensification [27–29]; and sustainable land management [17,30–32]. Hence, CSA is a suite of practices and technologies integrated into an agricultural system, often at different scales, rather than a specific, plot-based practice or technology [19]. Hence, CSA is sustainable agriculture that meets the needs of the present generation by ensuring food security and reducing agricultural livelihood vulnerability to climate change. It also meets the need of future generations by building climate resilience and reducing GHG emissions at the farm level for food production and the maintenance of ecosystem services.

Ethiopia has adopted CSA to meet the adaptation and mitigation objectives of the Climate-Resilient Green Economy (CRGE) strategy in the highland agriculture system [33]. The major initiatives are soil erosion or land degradation restoration practices through soil and water conservation measures, agroforestry, and area closures [33,34]. CSA practices in Ethiopia have been implemented primarily within integrated watershed management through projects such as the Sustainable Land Management Programme [33].

Despite the existence of literature on the effect of CSA practices on food security and welfare [35,36], literature on the synergy and trade-off effect of CSA innovations on food security, climate change adaptation, and mitigation benefit is scarce. [37,38] examined CSA innovations adoption and its effect on household income and resilience, yet their studies have limited insight into the synergy and trade-off effect. We hypothesized that multiple CSA innovations improve synergy among CSA goals and reduce trade-offs by ensuring food security, adaptation (reducing livelihood vulnerability to climate change), and mitigation (reducing GHG emissions at the farm level). Therefore, this study examined the effect of CSA innovations on agricultural sustainability in relation to improving synergy and reducing trade-offs among smallholder farmers in the Blue Nile highlands of Ethiopia.

## 2. Methodology

### 2.1. Study Area

The Choke mountain watershed is located in Ethiopia's Blue Nile highlands. It is located between 9°38′00″ to 10°55′24″ north latitude and 37°07′00″ to 38°17′00″ east longitude (Figure 1). It is located at an elevation of 2100 to 4113 m above sea level, and the total land surface area of the watershed is approximately 15,950 km$^2$, with an average annual rainfall of 200 to 2200 mm as well as an average annual temperature of 11.5 °C to 27.5 °C. The watershed has a slope gradient from flat to steep, and eight dominant soil types are found: Alisols, Andosols, Cambisols, Leptosols, Luvisols, Nitosols, Phaeozems, and Vertisols. The climate of the watershed ranges from the hot, arid climate of the Abay (Blue Nile) gorge to the cold and moist climate of the peak of Choke mountain [39].

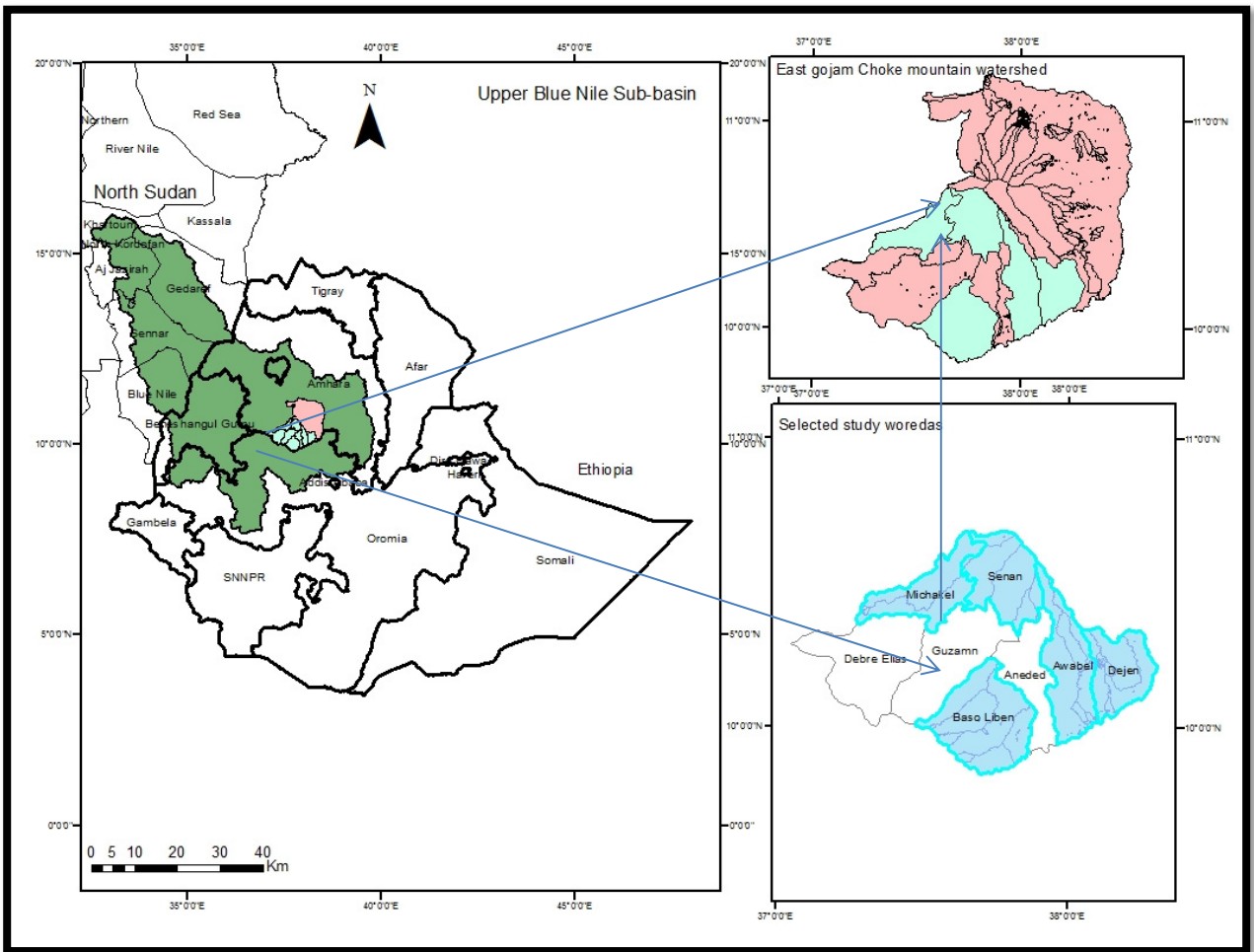

**Figure 1.** Map of the study area.

The study employed a mixed research method that uses both qualitative and quantitative data. The quantitative data were generated from household surveys. The household and plot-level survey data gathered consisted of information on socio-demographic and economic characteristics; climate-smart agricultural practices and preferences; crop production; livelihood assets; climate change risk and exposure; food consumption and frequency; and climate resilience. The qualitative data largely depended on information obtained through key informant interviews, observation, and document review. Key informant interviews were particularly used to generate in-depth information pertinent to climate trends over time, agricultural technologies, perception towards the technologies, major climate-related hazards, land use and land cover change, climate adaption, and mitigation measures. For these purposes, a total of 10 key informant interviews, 2 key informants per woreda, among woreda agricultural and natural resource officials were conducted. Five in-depth interviews of smallholder farmers in each woreda were conducted. The interviewee was selected based on age, community status, and engagement with government-led land restoration activities. In addition, secondary data were also collected from woreda agricultural offices through a desk review that included information on the agroecosystems, precipitation, temperature, land use, crop production, livestock, population, and other data related to the study objectives. The sample size was determined based on the formula obtained from [40], with the assumption that half of the population in the study area had adopted CSA innovations. Thus, we used the formula:

$$n_1 = \frac{Z^2_{1-\frac{\alpha}{2}}}{d^2} P(1-P) = \frac{(1.96)^2 (0.5)^2}{(0.05)^2} = 385$$

where $n_1$ is the initial sample size; $Z_{1-\alpha/2} = 1.96$ for a 95 percent confidence interval; $P$ is the proportion of smallholder farmers who adopt multiple CSA innovations, $P = 0.5$; and d is the error of margin, taking $d = 0.05$. There was a 10 percent non-response rate, which was 39 households. The sample size became 424 smallholder households.

A multi-stage sampling technique was used to select 424 households at random from the five woredas. The woredas were chosen through purposive sampling in the first stage based on the agroecosystem zone they represent. In the second stage, one kebele from each woreda was chosen at random. Gelegele from Dejen, Enebi from Awobel, Limichim from Basoliben, Debere klemu from Machakel, and Yeted from Sinan were among these kebeles. Finally, using a sampling frame of a one-to-five community mobilization group register, a systematic random sampling technique was used to select 424 households and 1818 farm plots based on probability proportional to size (PPS) (Table 1).

**Table 1.** Sample woredas/districts and kebeles.

| District/Woreda | Kebele | Sample Size | Agroecosystem Zone (AESZ) |
|---|---|---|---|
| Dejen | Gelgele | 77 | AESZ1: Lowland agroecosystem |
| Awabel | Enebi | 55 | AESZ2: Midland with black soil |
| Basoliben | Limichim | 104 | AESZ3: Midland with brown soil |
| Machakel | Debre Kelemu | 63 | AESZ4: Midland with sloping land |
| Sinan | Yeted | 125 | AESZ5: The hilly and mountainous highland |
| Total | | 424 | |

A structured household survey questionnaire focusing on household and farm characteristics was used for one-on-one interviews carried out using Android tablets by well-trained and experienced enumerators.

*2.2. Measurements*

In measuring the CSA triple benefit, food security, livelihood vulnerability, and carbon balance outcomes, the study used three standard food security, livelihood vulnerability, and carbon balance calculators. For food security measurement, chosen from among many food security measurements, the study used the food consumption score (FCS), which shows the quantity and quality of food consumed by a household in the last seven days [41]. FCS encompasses food security's food availability, access, and utilization aspects [42,43]. It is a composite score based on the dietary diversity, food frequency, and relative nutritional importance of the various food groups consumed. FCS is calculated as follows:

$$FCS = a_{staple}X_{staple} + a_{pulse}X_{Pulse} + a_{veg}X_{veg} + a_{fruit}X_{fruit} + a_{animal}X_{animal} + a_{sugar}X_{sugar} + a_{dairy}X_{dairy} + a_{oil}X_{oil} \quad (1)$$

where FCS is the food consumption score; $X_i$ is the frequencies of food items $i$ consumed in the past 7 days; $a_i$ is the weight of each food group; and $i$ is staple, pulse, vegetable, fruit, animal, sugar, dairy, and oil.

The thresholds for identifying households with poor and borderline food consumption were determined using dietary pattern assumptions, with a score of less than 21 showing a less than 1470 calorie intake (Kcal/capita/day) [44], or less than 1680 calorie intake (Kcal/capita/day) [45], and scores above 35 showing an acceptable threshold level of above 2100 calorie intake (Kcal/capita/day) [41,44]. Households with scores between 21 and 35 were considered to have borderline food consumption [46], a calorie intake of between 1470 and 2100 calories (Kcal/capita/day) [44] or a calorie intake between 1680 and 2100 calories (Kcal/capita/day) [45]. When compared to the calorie intake method, poor and borderline groups were considered to be food insecure [44].

Food availability was taken care of by the FCS [47]. As high food consumption increases the possibility that a household achieves nutrient adequacy, food utilization is included through the household's frequency of the utilization of the available food or accessed food [42,43,48]. A modified Household Dietary Diversity Score (HDDS) was calculated for each household using data on the consumption of different food items over

the previous 7 days [49]. The food items were categorized into 8 different food groups, with each food group counting toward the household score if a food item from the group was consumed by anyone in the household in the previous seven days. The modified HDDS, then, was a continuous score from 0 to 8. The food groups used to calculate the modified HDDS were: cereals, roots and tubers, pulses, vegetables, fruits, meat, eggs, fish, milk and milk products, oils and fats, sugar, and condiments.

A risk and hazard model was used in the study to assess smallholder farmers' household vulnerability to climate change [50–52]. Thus, the current study used an indicator (composite index) approach to assess smallholder farmers' vulnerability to climate change based on the livelihood vulnerability index (LVI) developed by [53] and adopted for agroecosystem analysis by [9]. However, unlike [9,53], this study used household-level primary data and took into account several variables to capture the level of exposure to climate hazards as well as the adaptive capacity and sensitivity to climate change. Hence, the indices used in this study integrated the LVI and the IPCC-LVI [9,53–56].

The present study focused only on major factors (major components and sub-components). Additionally, the study adopted an end-point vulnerability assessment framework that integrates vulnerability concepts that combine information on potential climate impacts and the socio-economic capacity to cope and adapt. It weighed the importance of various indicators and computed the LVI using the balanced weighting approach. The current vulnerability analysis involved the calculation of a balanced, weighted average LVI (composite index) in which each major component contributes equally to the overall index. Accordingly, first, the raw data were transformed into appropriate measurement units, such as percentages, ratios, and indices, and then the indicators measured on different scales were standardized (Equations (2) and (3)).

$$S_{xi} = \frac{S_{xa} - S_{xmin}}{S_{xmax} - S_{xmin}} \qquad (2)$$

while

$$S_{xi} = 1 - \frac{S_{xa} - S_{xmin}}{S_{xmax} - S_{xmin}} \qquad (3)$$

where $S_{xi}$ is the standardized value for the indicator $x$, $S_{xa}$ is the observed (average) major component indicator for CSA innovation $a$, and $S_{xmin}$ and $S_{xmax}$ are the minimum and maximum values, respectively, for the indicator across the seven CSA innovations. Then, the major component indicators are averaged (Equation (4)).

$$M_{aj} = \frac{\sum_{i=1}^{n} S_{xi}}{n} \qquad (4)$$

where $M_{aj}$ represents the **j** major components for CSA innovation a; index $S_{xi}$ represents the indicators for the major component $M$ indexed by $j$; $N$ runs from 1 to 9, which makes up each major component; and $n$ is the number of indicators in the major component $M$. Equation (5) combines the weighted averages of all the major components $M$ to generate the *LVI* score. The number of indicators which are compressed to determine the weights of each major component is $W_j$. Values for each of the eight major components for a CSA innovation $a$ were calculated and averaged (Equation (5)) to obtain *LVI*:

$$LVI_a = \frac{\sum_{j=1}^{N} W_j M_{aj}}{\sum_{j=1}^{N} W_j} \qquad (5)$$

where *LVIa* is the *LVI* for CSA innovation $a$, the value of which ranges from 0 (least vulnerable) to 1 (most vulnerable); $M_{aj}$ is the major component of CSA innovation $a$; and $W_j$ is the weight of the major component. The weights of each major component, $W_j$, are determined by the number of sub-components that make up each component, and all components contribute equally to the overall *LVI*. However, *IPCC-LVI* was calculated using the *IPCC* vulnerability definition. Major components were categorized into the three *IPCC* contributing factors of exposure (natural hazard and climate change), sensitivity

(ecosystem and agriculture), and adaptive capacity (wealth, innovation, infrastructure, and social network). Then, the *IPCC* contributing factors for CSA innovations were calculated based on the following formula (Equation (6)):

$$IPCC\_CF_a = \frac{\sum_{k=1}^{N} W_j M_{ak}}{\sum_{k=1}^{N} W_k} \tag{6}$$

where *IPCC_CF$_a$* denotes the *IPCC* contributing factor for CSA innovation *a*; $M_{ak}$ denotes the major component index by *k*; and *N* denotes the number of contributing LVI major components. The *IPCC-LVI* was calculated after the exposure, sensitivity, and adaptive capacity of CSA innovations. Hence:

$$LVI - IPCC = (E - AC) * S \tag{7}$$

where *IPCC-LVI* is the *LVI* value calculated following the *IPCC* vulnerability definition, which ranges from −1 (least vulnerable) to 1 (most vulnerable); *E* denotes exposure; *AC* denotes adaptive capacity; and *S* denotes sensitivity.

Following the guidelines of GHG calculator selection [57], the study selected the EX-ACT (EX-Ante Carbon Balance Tool) for the availability of data and simplicity of calculation [58]. The EX-ACT is an open-source GHG calculator consisting of a set of 18 modules linked to Microsoft Excel sheets, into which researchers insert information on the country- or region-specific data on soil types and climatic conditions together with basic data on land use, land use change, and land management practices foreseen under intervention activities as compared to a "business-as-usual" scenario [58]. Using EX-ACT, the GHG influxes from CSA innovations were calculated using Equations (8)–(10). This net emissions value per unit of farmland was considered as the carbon balance, i.e., net carbon emissions per hectare (ton $CO_2$ e/ha/yr). Based on [59], GHG emissions, carbon stock, and carbon balance at the farm level were calculated as follows:

$$\text{GHG emissions} = \text{activity data} * \text{GHG emission factor} \tag{8}$$

$$\text{Carbon stock} = \text{Above and below} - \text{ground biomass} * \text{carbon stock exchange factor} \tag{9}$$

$$\text{Carbon balance} = \sum \text{GHG emissions} - \sum \text{carbon stocks} \tag{10}$$

where activity data is GHG influxes, which include afforestation; annual agriculture; perennial agriculture; livestock; and input and investment. Hence, the GHG influxes were measured using land use/land cover change proxy measures such as the planting of trees such as Eucalyptus globulus, the annual crop land system that includes the cropping system, improved agronomic practices, improved nutrient management, no-till and residue retention, water management, manure application, yield per hectare, the number of livestock, the types of livestock, and the quantity of livestock products, liming, fertilizer (UREA and UPS), and energy consumption, i.e., wood fuel. These variables were used in the analysis and describe the GHG emissions from each CSA innovation intervention area that was measured using farm size (in ha), crop yield (qt/ha), the absolute number and types of livestock and livestock products (milk and meat in tons), lime use (in tons), and fire wood (in tons/year).

The data were subjected to descriptive statistics analysis to obtain frequencies and cross-tabulations, and the mean, standard deviation, and percentage were utilized depending on the nature of the variable and the need for presentation. STATA version 15 was used to analyze the data. T-test and chi-square test were used to determine whether the effects of variations in the CSA innovation adoption on food security or livelihood vulnerability or GHG emissions in the research area were statistically significant. The study used endogenous switching regression (ESR) to examine the impact and determinants of CSA innovations on livelihood vulnerability, food security, and GHG emission. The dependent variables for the ESR were the adopted CSA innovations such as improved variety, crop residue management, crop rotation, compost, row planting, soil and water

conservation, and agroforestry [60–66]. The independent variables were selected based on an extensive literature review on CSA innovations in sub-Saharan Africa and Ethiopia that was primarily based on past empirical literature on the determinants of food security or adaptation strategies or GHG emissions.

The adoption and impact of CSA innovations on food security, livelihood vulnerability, and GHG emissions were modeled in the setting of a two-stage framework using an endogenous switching regression model [67]. In the first stage, we used a selection model for CSA innovation adoption where a smallholder farmer chooses to implement CSA innovations if their gain with regard to food security, livelihood vulnerability, and GHG emissions outweighs the loss incurred due to its adoption.

In the second stage, the model estimation simultaneously controlled the effect of factors on adoption decisions and their outcomes by estimating a simultaneous equations model of the adoption of CSA innovation and its impact on the outcome variable with ESR using the full information maximum likelihood (FIML) estimation method. Instrumental variables were used as selection instruments, not only those automatically generated by the non-linearity of the selection model of adoption but also other variables that directly affect the adoption of CSA innovations, though not their impact on the outcome variables.

In our case, the study used selection instruments in the outcome variable function of the variables related to wealth, education, access to extension, and awareness of CSA. The study established the admissibility of these instruments through a simple falsification test; if a variable is a valid selection instrument, it affects adoption decisions but not the outcome (food consumption score, livelihood vulnerability index, and farm GHG emission).

To account for selection biases, an ESR model was used for the outcome variables (food consumption score, livelihood vulnerability index, farm GHG emission, and climate resilience capacity index), in which farmers face two regimes: Regime 1, to adopt and Regime 2, not to adopt, defined as follows:

$$U_{1i} = X_i\beta_1 + \epsilon_{1i} \tag{11}$$

$$U_{2i} = X_i\beta_2 + \epsilon_{2i} \tag{12}$$

$$G_i^* = \partial(U_{1i} - U_{2i}) + Z_i\alpha + u_i \tag{13}$$

Here, $G_i^*$ is a latent variable that determines the utility obtained whether the household $i$ adopts a CSA innovation or not; $U_{ji}$ is the outcome variable value of a household $i$ who adopts CSA innovation and j = Regime 1 and Regime 2; and $Z_i$ is a vector of characteristics that influences the decision to adopt the innovation but not the outcome variable value. $X_i$ is a vector of household characteristics that are thought to influence the decision to adopt the innovation, $\beta_1$, $\beta_2$, and $\gamma$ are vectors of parameters, and $u_i$, $\epsilon_{1i}$, and $\epsilon_{2i}$ are the error terms.

The regression model coefficient of adoption, which measures the impact of adopting the innovation, should be random. However, in the case of the adoption of CSA innovations, farmers freely choose the particular CSA innovation they want to adopt with their consent. Hence, there is the problem of self-selection, which leads to selection bias. The decision to adopt a given innovation is likely to be affected by unobservable characteristics that may be correlated with the outcome variables (food consumption score, livelihood vulnerability index, farm GHG emissions, and climate resilience capacity index). Finally, the error terms in Equations (12)–(14) are assumed to have a trivariate normal distribution ($v, \epsilon_1, \epsilon_2) \sim N(0, \Sigma)$:

$$\Sigma = \begin{matrix} \sigma_v^2 & \sigma_{v1} & \sigma_{v2} \\ \sigma_{1v} & \sigma_1^2 & . \\ \sigma_{2v} & . & \sigma_2^2 \end{matrix}$$

where $\sigma_v^2$ is the variance in the adoption of Equation (13), which is equal to 1, since the coefficients are estimable only up to a scale factor; $\sigma_1^2$ and $\sigma_2^2$ are the variances of the error terms in the outcome variable of Equations (11) and (12); and $\sigma_{1v}$ and $\sigma_{2v}$ represent the

covariance of $v_i$ and $\varepsilon_{1i}$ and $\varepsilon_{2i}$. Since Equations (11) and (12) are not observed simultaneously, the covariance between $\varepsilon_{1i}$ and $\varepsilon_{2i}$ is not defined (reported as dots in the covariance matrix). An important implication of the error structure is that, because the error term of the selection in Equation (13) $u_i$ is correlated with the error terms of the outcome variable of Equations (11) and (12) ($\varepsilon_{1i}$ and $\varepsilon_{2i}$), the expected values of $\varepsilon_{1i}$ and $\varepsilon_{2i}$, conditional on the sample selection, are nonzero.

$$E = (\varepsilon_{1i}|G_i = 1) = \sigma_{1v} \frac{\phi(Z_i\alpha)}{\Phi(Z_i\alpha)} = \sigma_{1v}\lambda_{1i} \tag{14}$$

$$E = (\varepsilon_{2i}|G_i = 0) = \sigma_{1v} \frac{\phi(Z_i\alpha)}{1 - \Phi(Z_i\alpha)} = \sigma_{2v}\lambda_{2i} \tag{15}$$

where $\phi$ (.) is the standard normal probability density function, $\Phi$(.) the standard normal cumulative density function, $\lambda_{1i} = \frac{\phi(Z_i\alpha)}{\Phi(Z_i\alpha)}$, and $\lambda_{2i} = -\frac{\phi(Z_i\alpha)}{1 - \Phi(Z_i\alpha)}$. If the estimated covariances $\sigma_{1v}$ and $\sigma_{2v}$ are statistically significant, then the decision to adopt and the outcome variable are correlated, that is, evidence of endogenous switching is found and rejects the null hypothesis of the absence of sample selectivity bias. An efficient method to estimate endogenous switching regression models is full information maximum likelihood estimation [68]. The logarithmic likelihood function [69] given the previous assumptions regarding the distribution of the error terms is:

$$\ln L_i = \sum_{i=1}^{N} A_i \left[ lnln\, \phi\left(\frac{\epsilon_{1i}}{\sigma_1}\right) - lnln\, \sigma_1 + lnln\, \Phi(\theta_{1i}) \right] + (1 - A_i)\left[ lnln\, \phi\left(\frac{\epsilon_{2i}}{\sigma_2}\right) - lnln\, \sigma_2 + lnln\, (1 - \Phi(\theta_{2i})) \right] \tag{16}$$

where $\theta_{ji} = \frac{Z_i\alpha + \frac{\epsilon_{ij}}{\sigma_j}\rho_j}{\sqrt{(1-\rho_j^2)}}$, and $j = 1, 2$, with $\rho_j$ denoting the correlation coefficient between the error term $u_i$ of the CSA innovation adoption in Equation (13) and the error term $\varepsilon_{ji}$ of Equation (16), respectively. The ESR model can be used to compare the expected outcome variable of the farm households that adopt particular innovation (a) to the farm households that do not adopt (b) and to investigate the expected outcome variable result in the counterfactual hypothetical cases (c), in which the adopted farm households do not adopt and (d) the non-adoption farm household adopts (Table 2).

**Table 2.** Conditional expectations, treatment, and heterogeneity effects.

| | Adoption Decision | | Treatment Effect |
|---|---|---|---|
| | **To Adopt** | **Not to Adopt** | |
| Adopters | Y11 = $E(U_{1i}|G_i = 1)$ | Y21 = $E(U_{2i}|G_i = 1)$ | ATT = Y11 − Y21 |
| Non-adopters | Y10 = $E(U_{1i}|G_i = 0)$ | Y20 = $E(U_{2i}|G_i = 0)$ | ATU = Y10 − Y20 |
| Heterogeneity effects | H1 = Y11 − Y10 | H2 = Y21 − Y20 | TH = ATT − ATU |

Note: Y11 and Y20 represent observed expected CSA innovation adoption outcome variable; Y10 and Y21 represent counterfactual expected outcome variable $G_i = 1$ if farm households adopt CSA innovation; $G_i = 0$ if farm households do not adopt; $Y_{1i}$: the outcome variable if farm households adopt; $Y_{2i}$: outcome variable if farm households do not adopt; TT: the effect of adopting the innovation on the farm households that adopt the innovation; TU: the effect of adoption of the innovation on the untreated; H1: the effect of base heterogeneity for farm households that adopt the innovation; H2: the effect of base heterogeneity for farm households that do not adopt the innovation; TH = (H1 − H2), i.e., transitional heterogeneity.

(a)  $E(U_{1i}|G_i = 1) = X_{1i}\beta_1 + \sigma_{1v}\,\lambda_{1i}$;
(b)  $E(U_{2i}|G_i = 0) = X_{2i}\beta_2 + \sigma_{2v}\,\lambda_{2i}$;
(c)  $E(U_{2i}|G_i = 1) = X_{1i}\beta_2 + \sigma_{2v}\,\lambda_{1i}$;
(d)  $E(U_{1i}|G_i = 0) = X_{2i}\beta_1 + \sigma_{1v}\,\lambda_{2i}$.

Cases (a) and (b) along the diagonal of Table 2 represent the actual expectations observed in the sample. Cases (c) and (d) represent the counterfactual expected outcome

variable. In addition, the effect of the treatment "to adopt" on the treated (ATT), as the difference between (a) and (c), was calculated as:

$$\text{ATT} = E(U_{1i}|G_i = 1) - E(U_{2i}|G_i = 1) = X_{1i}(\beta_1 - \beta_2) + (\sigma_{1v} - \sigma_{2v})\lambda_{1i}$$

which represents the effect of the adoption of CSA innovations on the outcome variable result of the farm households that actually adopt a particular CSA technology [70]. Similarly, the effect of the treatment on the untreated (TU) for the farm households that actually do not adopt was calculated as the difference between (d) and (b):

$$\text{TU} = E(U_{1i}|G_i = 0) - E(U_{2i}|G_i = 0) = X_{2i}(\beta_1 - \beta_2) + (\sigma_{1v} - \sigma_{2v})\lambda 2_i$$

The expected outcomes described in Equations (a)–(d) can also be used to calculate the effects of heterogeneity. The effect of base heterogeneity [71] for the group of farm households that decides to adopt as the difference between (a) and (d) can be calculated as:

$$\text{H1} = E(U_{1i}|G_i = 1) - E(U_{1i}|G_i = 0) = (X_{1i} - X_{2i})\beta_{1i} + \sigma_{1v}\,(\lambda_{1i} - \lambda_{2i})$$

Similarly, for the group of farm households that decides not to adopt, the effect of base heterogeneity is the difference between (c) and (b):

$$\text{H2} = E(U_{2i}|G_i = 1) - E(U_{2i}|G_i = 0) = (X_{1i} - X_{2i})\beta_{2i} + \sigma_{2v}\,(\lambda_{1i} - \lambda_{2i})$$

Finally, the transitional heterogeneity (TH) was investigated, that is, whether the effect of adopting the innovation is larger or smaller for farm households that actually adopt the innovation or for farm households that actually do not adopt; however, in the counterfactual case, if they do adopt, that is the difference between Equations (9) and (10) (i.e., TT and TU).

## 3. Results and Discussion

### 3.1. Livelihood Vulnerability and Adoption of CSA Innovations

The livelihood vulnerability index (LVI) provides information on the components of livelihood vulnerability. These components include natural disaster and climate change exposure, ecosystem and agriculture sensitivity, wealth, technology, infrastructure use, knowledge, and social networks. On the other hand, the IPCC-LVI identifies which of the exposure, adaptive capacity, and sensitivity factors has a significant impact on agricultural vulnerability to climate change.

The mean comparison of adopters and non-adopters of CSA innovations using the major components of exposure, sensitivity, and adaptive capacity is presented in Appendix A Table A1. Accordingly, adopters of improved varieties have a significantly lower natural disaster and climate change index and a higher ecosystem index but a lower agriculture index than non-adopters. Hence, adopters of improved varieties have a lower climate change exposure index than non-adopters due to their adoption of high-yielding and drought-resistant varieties [72,73]. In terms of adaptive capacity, adopters of improved varieties have significantly higher wealth and technology indices while having a lower knowledge skill index than non-adopters. Hence, adopters of improved varieties have a higher adaptive capacity index than non-adopters because the adoption of improved maize varieties increases the adoption of mineral fertilizers coupled with increased extension services [64,65]. Thus, adopters of improved varieties have a lower climate change exposure index as well as a higher adaptive capacity index than non-adopters. Consequently, their IPCC-LVI is closer to zero (0.09), which is significantly lower than that of non-adopters (0.12) ($p < 0.01$), which shows that adopters are significantly less vulnerable to climate change than non-adopters. Hence, the adoption of improved varieties significantly enhances adaptation to climate change, which concurs with studies [37,72,74], which reported that improved varieties enhance climate resilience and food security through the adop-

tion of high-yield, improved wheat varieties and drought-resistant maize varieties among smallholder farmers.

Adopters of crop residue management have a higher climate change exposure index than non-adopters. In terms of the sensitivity of their livelihood, adopters of crop residue management have a lower ecosystem and agriculture index than non-adopters, which results in them having a lower sensitivity index (0.36) than non-adopters (0.38) ($p < 0.01$). In terms of adaptive capacity, adopters have a significantly lower wealth index and a lower social network index than non-adopters ($p < 0.05$). Yet, adopters of crop residue management have a higher adaptive capacity index (0.35) than non-adopters (0.33), but there is no difference in the IPCC-LVI among adopters and non-adopters.

Crop residue management on farm plots might be considered essential in promoting the physical, chemical, and biological aspects of soil health in smallholder agriculture systems due to the dearth of substitute organic amendments. However, smallholder farmers must make trade-offs when managing crop residues because of several alternative uses such as livestock feed and fuel sources [35,75], which is the case in the study area. One reason for the trade-off effect of crop residue management is the lack of crop residue for farm management [76]. Furthermore, recent research has indicated that the quality of teff crop residue is the least desired benefit by farmers [66]. Thus, conclusions about the effect of crop residue management in reducing livelihood vulnerability to climate change in particular and enhancing adaptation to climate change in general are inconclusive.

Adopters of crop rotation have a higher natural disaster and climate change exposure index than non-adopters. This shows that areas with a variable climate or environmental stress-response characteristics adopt crop rotation, which supports the finding of [77], who reported that it allows farmers to grow products that can be harvested at different times and in different climates, helping weed management and reducing pests and diseases infestations [77]. In terms of adaptive capacity, the wealth index and knowledge index of adopters are significantly lower than non-adopters. However, the adaptive capacity index of adopters (0.35) is significantly higher than that of non-adopters (0.33), so technology, infrastructure, and social networks offset the deficits in the wealth and knowledge indices of adopters. However, the adaptive capacity index does not offset the natural disaster and climate change index. Hence, there is a significantly higher IPCC-LVI for adopters (0.12) than for non-adopters (0.10). Crop rotation makes farmers' livelihoods more vulnerable to climate change because farmers in spatiotemporally climate-change-exposed agroecosystems prefer crop rotation, and their adaptive capacities in terms of technology adoption, wealth, knowledge, infrastructure, and social networks do not offset exposure to climate change.

Among several problems with land degradation, low organic matter content is the major one [78]. Compost adopters have a higher ecosystem sensitivity index, which shows that compost is an important organic fertilizer because of its nutrient content and diverse effects on soil fertility and crop productivity, while non-adopters have a lower agricultural sensitivity index. This suggests that the adoption of compost improves soil structure, resulting in greater resistance to erosion, improved water infiltration, and increased water holding capacity, which results in improved crop yield [17]. Hence, adopters of compost have a significantly higher sensitivity index (0.38) than non-adopters (0.36) ($p < 0.01$) because of the higher ecosystem sensitivity of the agroecosystems. In terms of adaptive capacity, adopters of compost have significantly lower wealth and infrastructure indexes than non-adopters. However, the adaptive capacity index of adopters of compost (0.34) is significantly higher than non-adopters (0.32) ($p < 0.01$). Hence, there is a significantly lower IPCC-LVI for adopters (0.11) than for non-adopters (0.12) ($p < 0.01$). Thus, the adoption of compost significantly reduces livelihood vulnerability to climate change.

Farmers adopt row planting not only to reduce seed rates but also to allow more spacing between seedlings, to permit easy weeding, and to reduce competition between seedlings [79], which increases productivity and reduces vulnerability to climate change. The study shows that adopters of row planting have a lower natural disaster and climate

change exposure index than non-adopters. This suggests that adopters of row planting live in less climate-change-exposed agroecosystems than non-adopters. Adopters also have a higher ecosystem sensitivity index but a lower agricultural sensitivity index than non-adopters. This suggests that the higher ecosystem sensitivity of adopters of row planting is offset by good crop yield. However, due to the higher ecosystem sensitivity of the agroecosystem, adopters of row planting have a significantly higher sensitivity index (0.38) to climate change than non-adopters (0.34) ($p < 0.01$). In terms of adaptive capacity, though adopters of row planting have significantly lower wealth and knowledge than non-adopters, the overall adaptive capacity index of adopters of row planting is higher (0.34) than that of non-adopters (0.32) ($p < 0.01$). The IPCC-LVI result, on the other hand, reveals that the agricultural livelihood vulnerability to climate change of adopters of row planting (0.106) is significantly lower than that of non-adopters (0.123) ($p < 0.01$). Hence, the adoption of row planting has the significant impact of reducing agricultural livelihood vulnerability to climate change and enhances adaptation to climate change in smallholder agriculture systems. This result concurs with studies such as [80] which reported that row planting enhances food security and adaptation to climate change.

Soil and water conservation (SWC) techniques include soil bunds, stone bunds, bench terraces, vegetative barriers, and tied ridges [81]. SWC adopters have significantly higher natural disaster and climate change exposure indexes than non-adopters. This shows that adopters of SWC are farmers who live in the lowland Abay gorge (AESZ1), the midland sloping land (AESZ4), and the hilly and mountainous highland (AESZ5) agroecosystem zones. Hence, the climate change exposure index of adopters (0.67) of SWC is significantly higher than that of non-adopters (0.59) ($p < 0.01$). Although they have a marginally significantly higher ecosystem index (0.32) than non-adopters (0.31) ($p < 0.1$), there is no significant sensitivity index difference between adopters and non-adopters of SWC. In terms of adaptive capacity, the innovation index of adopters of SWC is lower than that of non-adopters. Hence, the IPCC-LVI result shows that adopters of SWC have higher agricultural livelihood vulnerability to the climate change index (0.123) than non-adopters (0.097) ($p < 0.05$). Thus, adopters of SWC are more vulnerable to climate change than non-adopters. This result concurs with the finding that the effect of SWC on improving crop yield is dependent on rainfall characteristics and types of crop, slope, and soil. Moreover, the effect of SWC on crop yield is negatively correlated with rainfall for SWC techniques, including level Fanya juu, graded soil bunds, stone bunds, and trash lines, which are the main characteristics of the Choke mountain watershed [81,82].

Agroforestry is a climate-smart agriculture system that diversifies the environmental and socioeconomic benefits of smallholder farmers sustainably, and agroforestry adopters are thought to be more resilient to the increased intensity of extreme weather events caused by climate change [23]. In this study, agroforestry adopters have a higher (0.67) climate change exposure index than non-adopters (0.62) ($p < 0.05$). In terms of the sensitivity index, they have a higher ecosystem index (0.35) and a lower agricultural sensitivity index (0.30) than non-adopters. However, they have a significantly higher sensitivity index (0.384) than non-adopters (0.365) ($p < 0.05$). In terms of adaptive capacity, they have higher technology adoption (0.83) and lower wealth (0.61) and knowledge (0.75) indexes than non-adopters. However, adopters of agroforestry have a significantly higher adaptive capacity (0.35) index than non-adopters (0.33) ($p < 0.01$). Unfortunately, the IPCC-LVI result shows that there is no significant difference between the livelihood vulnerability index of adopters and non-adopters of agroforestry (Appendix A Table A1). This is due to the marginal adoption of agroforestry as farmland border trees and homestead gardens, as well as wood lots, rather than as an agricultural system that supports their livelihood.

### 3.2. Food Security and Adoption of CSA Innovations

From the total sample households, less than one-third (33 percent) of the households were food secure (had an acceptable food consumption score), whereas more than two-

thirds of the households were food insecure, with 46 and 21 percent of the households having a borderline and poor food consumption score, respectively (Figure 2).

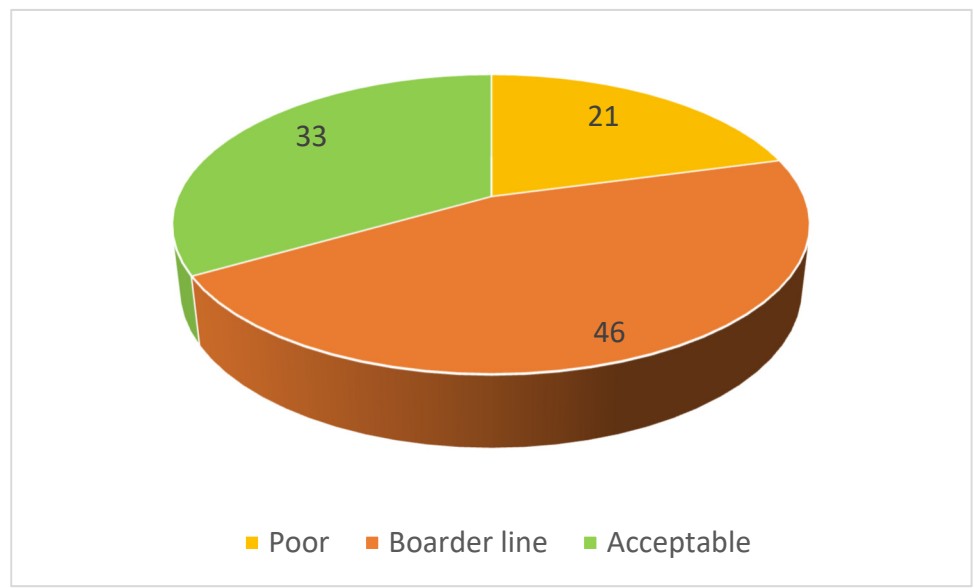

**Figure 2.** Household food security status.

Smallholder farmers adopt CSA innovations for their different CSA benefits, either for food security or climate adaptation or both. Table 3 shows that crop residue management (positive) and SWC (negative) adoption have a significant relationship with the food security status of the household ($p < 0.01$). Adopters of SWC have more poor food consumption households than non-adopters, indicating that more SWC adopter households are food insecure than non-adopter households, i.e., they lack daily staple and vegetable consumption, as well as consumption of oil and pulses four times a week, whereas adopters of crop residue management have more acceptable food consumption households than non-adopters, indicating that they represent food-secure households with daily staple and vegetable consumption, as well as consumption of oil and pulses at least four times a week.

**Table 3.** Relationship between food security and adoption of CSA innovation.

| CSA Innovation | Adoption Category | Food Security Status | | | Chi$^2$ |
| | | Poor | Borderline | Acceptable | |
| --- | --- | --- | --- | --- | --- |
| Improved variety | Adopter | 17.8 | 43.4 | 38.8 | 2.6 |
| | Non-adopter | 22.1 | 46.9 | 31.0 | |
| Crop residue management | Adopter | 9.7 | 44.1 | 46.2 | 39.4 *** |
| | Non-adopter | 30.3 | 47.4 | 22.4 | |
| Crop rotation | Adopter | 22.6 | 47.1 | 30.3 | 1.1 |
| | Non-adopter | 19.8 | 45.2 | 35.1 | |
| Compost | Adopter | 21.9 | 44.4 | 33.7 | 0.83 |
| | Non-adopter | 18.8 | 48.6 | 32.6 | |
| Row planting | Adopter | 18.9 | 46.3 | 34.8 | 3.14 |
| | Non-adopter | 26.7 | 44.6 | 28.7 | |
| Soil and water conservation (SWC) | Adopter | 30.8 | 42.5 | 26.6 | 27.9 *** |
| | Non-adopter | 10.5 | 49.3 | 40.2 | |
| Agroforestry | Adopters | 17.2 | 49.4 | 33.3 | 0.97 |
| | Non-adopters | 21.7 | 44.9 | 33.3 | |

Significance level: *** $p < 0.01$.

In terms of HDDS, adopters of SWC have the lowest (3.83), while adopters of agroforestry have the highest HDDS (4.43). Adopters of crop rotation, compost, crop residue management, and row planting have a medium HDDS value, which shows that the dietary diversity of adopters of these CSA innovations still needs improvement. For the diversity and quality of food consumed in the last seven days, the seven-day-recall household

dietary diversity score (HDDS) was compared among the adopters and non-adopters of specific CSA innovations. Hence, adopters of improved varieties (4.25) have a higher HDDS than non-adopters (3.78); adopters of crop residue management (4.11) have a higher HDDS than non-adopters (3.76); adopters of compost (4.07) have a higher HDDS than non-adopters (3.60); adopters of row planting (4.06) have a higher HDDS than non-adopters (3.45); and adopters of agroforestry (4.43) have a higher HDDS than non-adopters (3.80). Thus, adopters of improved varieties, crop residue management, compost, row planting, and agroforestry consume better-diversified diets of food than non-adopters (Figure 3). Therefore, the adoption of these CSA innovations improves the food utilization component of the food security of smallholder households. However, the effect of the adoption of CSA innovation differs for different food security components. For instance, improved variety, crop residue management, compost, and row planting have significant positive effects on food availability and access to food security, while agroforestry has a significant positive effect on food utilization of the food security component. Moreover, SWC negatively affects the food availability and access component of food security.

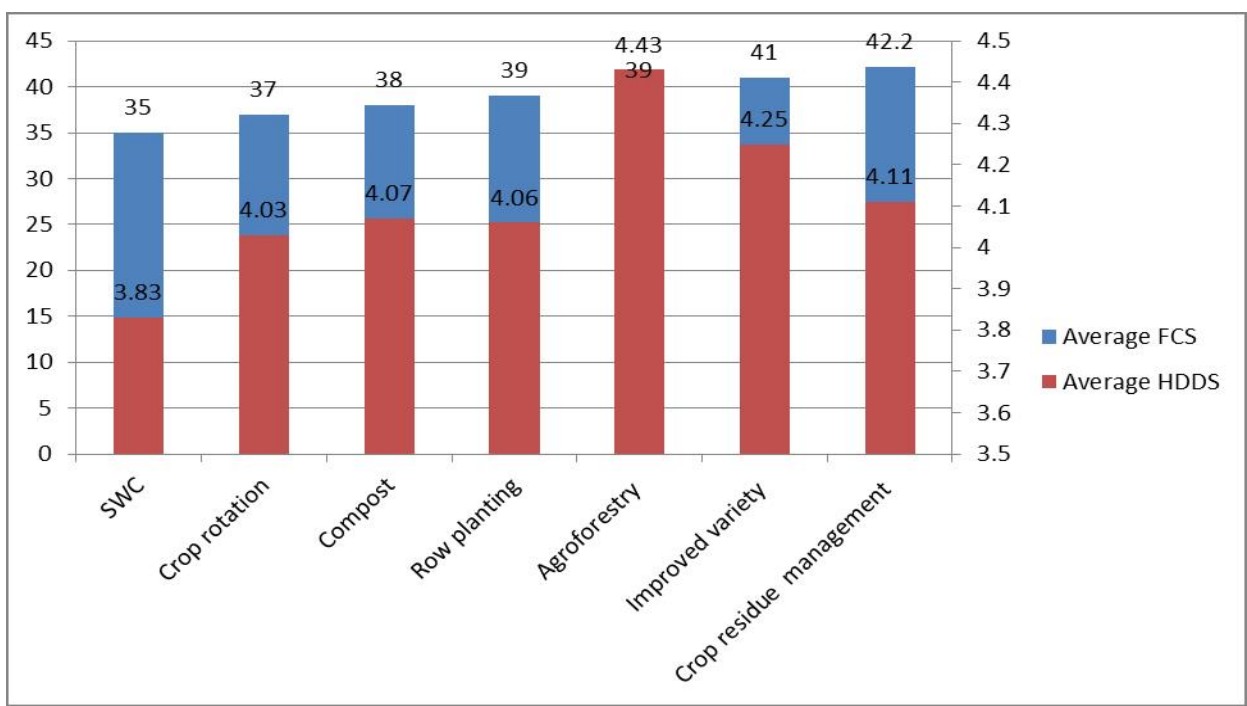

**Figure 3.** Food security of adopters of CSA innovations.

To reveal the effect of CSA adoption on the FCS, a simple mean comparison, along with an independent *t*-test, was conducted between the adopters and non-adopters of each CSA innovation. Table 4 shows that both adopters of improved varieties (40.6) and non-adopters (37.1) have an acceptable food security status, with adopters having a significantly higher FCS value than non-adopters with a positive effect of food security ($p < 0.01$), which concurs with studies [38,62], which reported that improved varieties significantly enhance household food security through the adoption of high-yield, improved wheat varieties. Similarly, both adopters of row planting (38.9) and non-adopters (36.2) have an acceptable food security status with a positive effect on food security ($p < 0.05$), which concurs with the finding of [35], who reported that row planting enhances the food security of adopters. Adopters of crop residue management have acceptable food security (42.2) and a significantly higher FCS value than non-adopters, who have borderline food security (34.7) ($p < 0.01$). Likewise, adopters of SWC have borderline food security (35.0) and a significantly lower FCS value than non-adopters who have acceptable food security (41.0), with a negative effect on food security ($p < 0.01$). This result concurs with the finding

that the effect of SWC on improving crop yield was dependent on rainfall characteristics, type of crop, slope, and soil. Moreover, the effect of SWC on crop yield was negatively correlated with rainfall for SWC techniques, including level Fanya juu, graded soil bunds, stone bunds, and trash lines, which are the main characteristics of the Choke mountain watershed [81,82]. In general, improved varieties, row planting, and crop residue management enhance food availability as well as the food access component of food security for smallholder households, whereas, SWC reduces the food availability as well as the food access component of food security for smallholder households.

**Table 4.** Comparison of food consumption score (FCS).

| CSA Innovations | Food Consumption Score (FCS) | | | Household Dietary Diversity Score (HDDS) | | |
|---|---|---|---|---|---|---|
| | Adopter | Non-Adopter | *t* Value | Adopter | Non-Adopter | *t* Value |
| Improved varieties | 40.6 (1.0) | 37.1 (0.6) | 3.0 *** | 4.25 | 3.78 | 0.47 *** |
| Crop residue management | 42.2 (0.7) | 34.7 (0.7) | 7.3 *** | 4.11 | 3.76 | 0.35 *** |
| Crop rotation | 37.4 (0.9) | 38.6 (0.7) | 1.0 | 4.03 | 3.87 | 0.17 |
| Compost | 38.4 (0.7) | 37.8 (0.9) | 0.6 | 4.07 | 3.60 | 0.49 *** |
| Row planting | 38.9 (0.6) | 36.2 (1.1) | 2.1 ** | 4.06 | 3.45 | 0.58 *** |
| SWC | 35.4 (0.8) | 41.0 (0.7) | −5.3 *** | 3.83 | 4.01 | 0.18 |
| Agroforestry | 38.9 (1.1) | 38.0 (0.6) | 0.7 | 4.43 | 3.80 | 0.63 *** |

Standard errors in parentheses: ** $p < 0.05$, *** $p < 0.01$.

### 3.3. GHG Emissions and Adoption of CSA Innovations

The descriptive statistics were calculated for the GHG sink as well as the GHG source among CSA innovations. In the analysis, the five GHG influxes were calculated separately and summarized. These GHG influxes were afforestation; annual agriculture; perennial agriculture; livestock; input and investment. Hence, the GHG influxes were measured using land use/land cover change proxy measures such as the planting of trees such as *Eucalyptus globulus*, the annual crop land system, which includes the cropping system, improved agronomic practices, improved nutrient management, no-till and residue retention, water management, manure application, yield per hectare, the number of livestock, the types of livestock, and the quantity of livestock products, liming, and fertilizer (UREA and UPS), and energy consumption, i.e., consumption of wood fuel. These variables were used in the analysis and describe the GHG emissions from each CSA innovation intervention area that was measured using farm size (in ha), crop yield (qt/ha), the absolute number and types of livestock, number of livestock products (milk and meat in tons), amount of lime use (in tons), and amount of fire wood used (in tons/year) (Appendix A Table A2).

In terms of afforestation (wood lot), adopters of compost afforest the largest area (29 ha) with Eucalyptus wood lot, while adopters of agroforestry afforest the smallest area (12 ha). This is because farmers allocate their land for home gardens, and there is no extra area for wood lot available. The productivity of each crop shows that adopters of row planting have the highest yield for teff (47.8 qt/ha), maize (44.5 qt/ha), wheat (47.7 qt/ha), potato (18.8 qt/ha), faba bean (9.1 qt/ha), and barley (9.3 qt/ha), while adopters of agroforestry have the lowest productivity for teff (13.6 qt/ha), maize (14.6 qt/ha), wheat (16.5 qt/ha), potato (6.1 qt/ha), faba bean (2.6 qt/ha), and barley (2.9 qt/ha). Livestock holding shows adopters of row planting have the highest number of livestock and types among CSA innovation adopters. In terms of farm output (milk and meat), adopters of row planting (59 tons) have the highest output, while adopters of agroforestry have the lowest farm output. This is due to the marginal adoption of agroforestry as farmland border trees and homestead gardens, as well as wood lots, rather than as an agricultural system that supports their livelihood.

Input and investment include the use of chemical fertilizer. Adopters of row planting have the highest input use among CSA innovations. Energy consumption shows adopters of row panting have the highest tonnage of firewood used in a year. Annual agriculture and afforestation or wood lots are the two important sinks of GHG emissions, while

livestock and input and investment are the two consistent sources of GHG emissions in the smallholder farming system. The C sequestration and GHG emission potential of each CSA innovation, however, differ, along with the GHG influxes. In terms of CSA innovations, compost (−10.36 MtCO2e) and row planting (−8.87 MtCO2e) have the highest, while crop residue management (−2.147 MtCO2e) has the least potential to sequester carbon dioxide equivalent GHG emissions in 20 years (Figure 4).

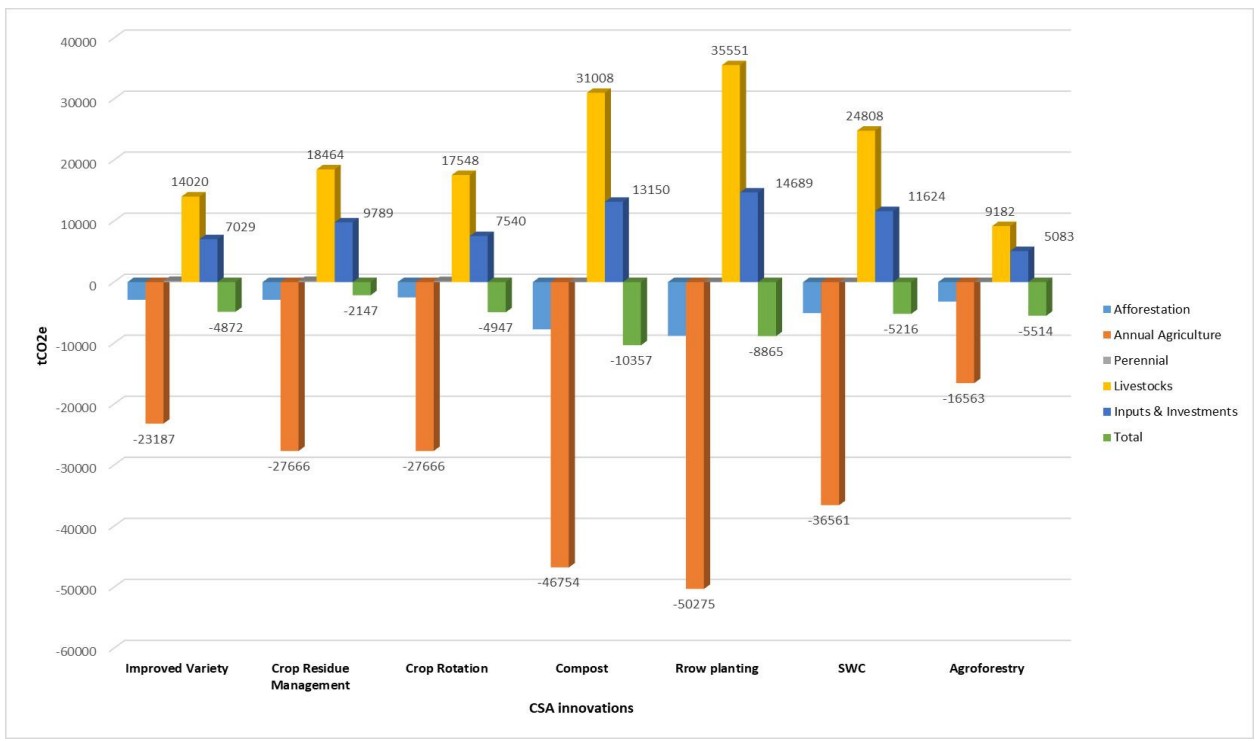

**Figure 4.** Gross influxes of GHG and adoption of CSA innovations.

Whole-farm GHG balances were disaggregated into farm compartments (Figure 5). In terms of biomass, all CSA innovations are sinks, and they have similar GHG sink potential. Soil has the highest GHG sink potential, with the highest sink potential recorded for compost and row planting adopters. The other source of GHG is enteric fermentation (livestock), which mainly emits CH4; in this regard, row planting and SWC adopters are the major emitters of GHG from enteric fermentation because they have a high density of livestock and have the largest enteric fermentation emissions with mean values of 2.4 tCO$_2$ eq ha$^{-1}$ yr$^{-1}$, while adopters of crop residue management have the lowest enteric fermentation emission (1.8 t CO$_2$ eq ha$^{-1}$ yr$^{-1}$). Carbon (CO$_2$) and nitrous oxide (N$_2$O) emissions from inorganic fertilizer (UREA and NPS) application accounts show that adopters of improved variety, row planting, SWC, and agroforestry are the biggest sources of GHG emission in terms of input use.

C sequestration in soils and biomass growth, when combined, offsets 5 to 52 percent of farm emissions. Woody biomass (above- and belowground) accounts for 5.1 percent of C removals among farm components affecting C sequestration. Farmers who use row planting and compost have the highest tree density and the highest C fluxes in above- and belowground biomass, with 0.89 t CO$_2$ eq ha$^{-1}$ yr$^{-1}$. The remaining 64–94% of total farm GHG removals (5.7 t CO$_2$ eq ha$^{-1}$ yr$^{-1}$) are accounted for by soil C sequestration. Compost users who apply the most manure to their soils experience soil C sequestration rates greater than 2.7 t CO$_2$ eq ha$^{-1}$ yr$^{-1}$. Among cropping systems, wheat–maize registers the highest rates of C sequestration (25% of soil removals), followed by teff–wheat (22%) and potato–barley (20%).

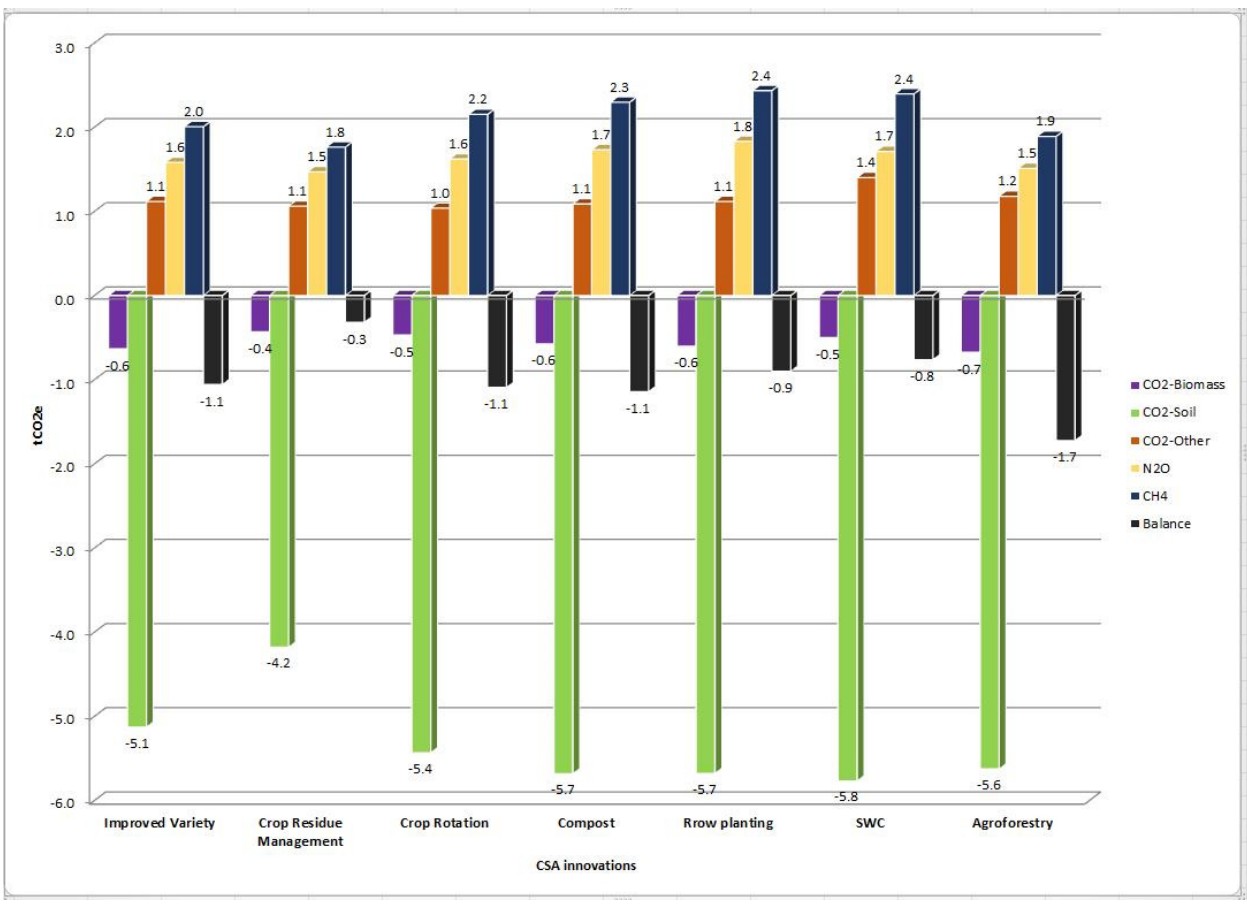

**Figure 5.** Comparison of GHG emission sources and CSA innovations.

Total emissions per year and intensity per output were calculated for farm-level GHG emissions. These two measurements show a difference in the levels of GHG emissions of CSA innovations for adopters. Hence, adopters of compost and SWC have lower farm GHG emissions per year than non-adopters. Similarly, crop residue management and compost adopters have a lower intensity of GHG emissions (net GHG emissions per yield) than non-adopters. Compost adopters have both lower gross emissions and net GHG emissions per yield than non-adopters. As a result, crop residue management, compost, and SWC are GHG emission sinks, and these CSA innovations can increase farm-level carbon sequestration capacity (Table 5).

**Table 5.** Comparison of GHG emissions among CSA innovations.

| CSA Innovations | GHG Emission (per ha per year) | | | Net GHG Emission (per ha per year per yield) | | |
| --- | --- | --- | --- | --- | --- | --- |
| | Adopter | Non-Adopter | *t* Value | Adopter | Non-Adopter | *t* Value |
| Improved varieties | 3.35(0.3) | 3.56(0.2) | −0.5 | 2.2(0.2) | 2.4(0.3) | −0.8 |
| Crop residue management | 3.21(0.3) | 3.75(0.2) | −1.44 | 1.9(0.2) | 2.7(0.4) | −1.7 * |
| Crop rotation | 3.12(0.3) | 3.7(0.2) | −1.5 | 2.0(0.3) | 2.5(0.3) | −1.0 |
| Compost | 3.0(0.2) | 4.41(0.3) | −3.5 *** | 1.8(0.2) | 3.2(0.5) | −2.9 *** |
| Row planting | 3.4(0.2) | 3.81(0.4) | 0.9 | 2.2(0.3) | 2.7(0.5) | −0.9 |
| SWC | 3.1(0.3) | 3.88(0.2) | −2.0 ** | 2.0(0.3) | 2.6(0.4) | −1.5 |
| Agroforestry | 3.4(0.5) | 3.5(0.2) | 0.3 | 2.4(0.3) | 2.3(0.3) | 0.2 |

Standard errors in parentheses; * $p < 0.1$, ** $p < 0.05$, *** $p < 0.01$.

### 3.4. Impacts of Climate-Smart Agriculture Innovations: ESR Estimation Results

Research on impact evaluation usually faces two main challenges: establishment of a viable counterfactual, attribution effect and addressing long, unpredictable lag times [41].

The sample mean comparison shows the impact of CSA innovation on the three CSA pillars. However, this naive comparison drives misleading conclusions because the approach does not consider that the difference in the outcome may be caused by observable as well as unobservable characteristics. Hence, the ESR model was used to prove that, indeed, the adopters and non-adopters of specific CSA innovation have different outcome values, taking into consideration the unobservable characteristics that may correlate. ESR was conducted in a two-stage process, where the first stage models the adoption equation, and the second stage models the effect of adoption on the outcome variable.

This paper focused on the impact of CSA innovations on the outcome variables of livelihood vulnerability, food security, and farm GHG emissions. Hence, Column A of Table 6 presents the true average adoption effects of CSA innovation on livelihood vulnerability, food security, and GHG emission under actual adoption conditions. Column B shows the counterfactual values of livelihood vulnerability, food security, and GHG emission. Column C presents the effect of CSA innovation on livelihood vulnerability, food security, and GHG emissions, computed as the difference between Columns A and B.

Table 5 shows that the adoption of improved varieties has lower livelihood vulnerability and GHG emission, with a higher food security status than actual as well as counterfactual non-adopters. Hence, the adoption of improved varieties significantly reduces agricultural livelihood vulnerability to climate change, enhances food security, and reduces GHG emissions. This finding concurs with [64,83–85], who reported that improved varieties not only enhance productivity but also increase income and may effectively reduce food insecurity and poverty in the medium-to-long term [86]. Moreover, the finding concurs with [20], which reported that adoption of improved variety, inter alia, reduces livelihood vulnerability and manages risk more effectively for smallholder farmers so that they can adapt to climate change and extreme weather events such as drought shock.

Adoption of crop residue management has higher livelihood vulnerability and higher GHG emissions, i.e., maladaptation, with higher food security than actual and counterfactual non-adopters. Adoption of crop residue management improves food security, concurrent with literature in Mozambique, which showed that crop residue management with associated practices has significantly improved food security in Zimbabwe and Malawi [86] but increased the agricultural livelihood vulnerability to climate change as well as, against all odds, increased the GHG emission from smallholder farms. This finding is in line with [87], who further reported that the adoption of crop residue management among other innovations not only enhances food security but also is an effective strategy for improving rural populations' well-being. However, the maladaptation of crop residue management comes from the competition of crop residue for livestock feed and fuel sources rather than soil fertility management.

Adoption of crop rotation has lower agricultural vulnerability to climate change, lower food security, i.e., maladaptation, and lower GHG emission than actual as well as counterfactual non-adopters. Hence, the adoption of crop rotation reduces the agricultural livelihood vulnerability to climate change and farm GHG emissions while reducing the food security of smallholder agriculture households, which is maladaptation. Previous findings [84] support this result, reporting that the adoption of crop rotation increases crop revenue per hectare. Moreover, the 31-year data on crop rotation data [88] supported this finding and reported that crop rotation, along with minimum tillage, increases yield in hot and dry years, which is a highly likely scenario under future changes. Furthermore, this finding is also consistent with [89], which reported that crop rotation not only reduces the risk of pests and disease, which is the main cause of crop failure and reduced yield in Africa due to projected climate change [2], but also suppresses weed infestation [90]. In addition, crop rotation and consumption of pulses are strongly directly correlated [91]. Furthermore, this finding supports the finding of [92], which reported that crop rotations increase soil organic carbon in smallholder agriculture.

**Table 6.** Average expected LVI, FCS, and GHG emission with adoption of CSA innovations.

| CSA Innovations | Livelihood Vulnerability Index (LVI) | | | Food Consumption Score (FCS) | | | Actual GHG If Farm Households Do Adopt (A) | Farm GHG Emission Counterfactual GHG if Farm Households Do Not Adopt (B) | Adoption Effects on GHG (C) |
|---|---|---|---|---|---|---|---|---|---|
| | Actual LVI If Farm Households Do Adopt (A) | Counterfactual LVI If Farm Households Do Not Adopt (B) | Adoption Effects on LVI (C) | Actual FCS If Farm Households Do Adopt (A) | Counterfactual FCS if Farm Households Do Not Adopt (B) | Adoption Effects on FCS (C) | | | |
| Improved variety | 0.65(0.003) | 0.69(0.002) | ATT = −0.04 (0.004) *** | 40.2(0.51) | 28.2(0.5) | ATT = 12.0 (0.7) *** | 3.65(0.15) | 5.6(0.21) | ATT = −193 (0.25) *** |
| Crop residue management | 0.67(0.002) | 0.63(0.002) | ATT = 0.034 (0.004) *** | 42.16(0.35) | 15.31(0.44) | ATT = 26.85(0.57) *** | 3.2(0.19) | −1.8(0.19) | ATT = 4.95 (0.27) *** |
| Crop rotation | 0.68(0.003) | 0.72(0.003) | ATT = −0.043 (0.004) *** | 37.3(0.42) | 38.75(0.38) | ATT = −1.41 (0.56) ** | 3.28(0.15) | 4.7(0.21) | ATT = −1.4 (0.26) *** |
| Compost | 0.67(0.002) | 0.71(0.002) | ATT = −0.05 (0.003) *** | 38.4(0.3) | 23.7(0.42) | ATT = 14.75(0.50) *** | 3.11(0.14) | 5.8(0.15) | ATT = −2.68 (0.21) *** |
| Row planting | 0.67(0.002) | 0.72(0.20) | ATT = −0.06 (0.003) *** | 38.9(0.24) | 37.1(0.31) | ATT = 1.77 (0.39) *** | 3.51(0.11) | 4.57(0.15) | ATT = −1.05 (0.19) *** |
| SWC | 0.68(0.002) | 0.72(0.002) | ATT = −0.05 (0.003) *** | 35.5(0.4) | 33.75(0.26) | ATT = 1.69 (0.52) *** | 3.05(0.22) | −2.14(0.13) | ATT = 5.2 (0.18) *** |
| Agroforestry | 0.67(0.005) | 0.73(0.003) | ATT = −0.06 (0.005) *** | 38.75(0.67) | 23.6(0.49) | ATT = 15.11(0.83) *** | 3.6(0.17) | 7.5(0.13) | ATT = −3.93 (0.36) *** |

Standard errors in parentheses ** $p < 0.05$ and *** $p < 0.01$.

Adoption of compost has lower agricultural vulnerability to climate change and GHG emissions but higher food security than actual as well as counterfactual non-adopters. Hence, the adoption of compost reduces the agricultural livelihood vulnerability to climate change and farm GHG emissions while enhancing the food security of smallholder agriculture households. This finding concurs with several pieces of literature which reported that manure, organic fertilizer, or compost enhances food security and reduces GHG emissions [93–95].

Adoption of row planting has lower agricultural vulnerability to climate change and GHG emissions but a higher FCS than non-adoption. Hence, the adoption of row planting enhances food security while reducing the agricultural livelihood vulnerability to climate change and GHG emissions from smallholder farms. Literature that examined the impact of row planting on labor productivity showed that row planting significantly increases the total labor requirement and allocation, resulting in a substantial drop in labor productivity [96]. Row planting increases not only teff yield but also teff income, as well as per capita food consumption [80,97–99].

Adoption of soil and water conservation measures such as soil/stone bunds has lower livelihood vulnerability but higher food security and GHG emission, i.e., maladaptation, than non-adoption. The adoption of SWC enhances food security and reduces agricultural livelihood vulnerability to climate change but increased smallholder farm GHG emissions. This finding backs up previous research that evaluated the impact of SWC practices in Ethiopia and found that SWC practices are effective in reducing surface run-off and nutrient loss, as well as controlling soil erosion [10,100,101]. However, studies have shown that the impacts of SWC practices on crop yield and the economic viability of SWC practices are inconsistent, and results are site specific [102]. Additionally, soil and stone bunds reduce crop yield for the first few years [103,104] while increasing crop yield [105]. Soil and water conservation improve crop productivity and crop yields on terraced fields for teff, barley, and maize [106]. Although the literature on the impact of SWC on food security is limited, a study in Eastern Ethiopia discovered that adopting soil and water conservation not only positively impacts per capita food consumption expenditure and net crop value, but also significantly reduces the likelihood of farmers being food insecure [107]. Adoption of agroforestry has a lower LVI and GHG emission but a higher FCS than actual as well as counterfactual non-adopters. Hence, the adoption of agroforestry reduces agricultural livelihood vulnerability to climate change and enhances food security while reducing smallholder farm GHG emissions.

Hence, a smallholder agriculture system can enhance food security, adapt to climate change, or build resilience and reduce GHG emissions by implementing a variety of CSA innovations such as sustainable land management (SLM), intercropping, crop rotation, soil and water conservation, modern input use, and agroforestry on farm plots.

## 4. Conclusions

Climate-smart agriculture (CSA) innovations are essential for enhancing household food security, reducing vulnerability, and reducing farm-level GHG emissions in the face of climate change for a sustainable agricultural system. However, the empirical foundation for understanding how farm households achieve these three major goals of the current climate and development agenda, or the "triple benefit", is far from being established. In this paper, the effects of CSA innovations on the triple benefit were examined. An endogenous switching regression model was used to explore a survey of 424 farm households and 1818 farm plots in the Blue Nile highlands of Ethiopia. Results showed that the adoption of improved variety, compost, row planting, and agroforestry simultaneously delivers the three CSA benefits of enhanced household food security, reduced vulnerability, and reduced farm-level GHG emissions. Similarly, the adoption of crop rotation reduces livelihood vulnerability and farm GHG emissions simultaneously. Soil and water conservation measures, or adoption of stone/soil bunds on farm plots, meet the goal of climate change adaptation as well as food security, as they enhance food security and reduce livelihood

vulnerability simultaneously, whereas, the adoption of crop residue management meets only the food security goals of CSA. Some of the CSA innovations, improved variety, compost, row planting, and agroforestry, provide farmers with the benefit of enhanced food security and climate change adaptation and reduced GHG emission from farm plots, while other CSA innovations, crop rotation and SWC, also deliver the two CSA pillars as they provide farmers with either enhanced food security and/or reduced livelihood vulnerability and/or reduced GHG emissions. Unfortunately, adopting crop residue management, one of the recommended CSA practices in Ethiopia, does not deliver at least two of the CSA pillars, which means it is not a CSA innovation. Farmers should be encouraged to adopt improved variety, crop rotation, compost, row planting, soil and water conservation, and agroforestry as the best portfolio of CSA innovation for highland smallholder agriculture systems as these innovations reduce the trade-off and increase synergy among CSA pillars and maintain agricultural sustainability in the face of climate change. Policies that encourage simultaneous adoption of the CSA portfolio should be devised as incentives. Competition surrounding crop residue utilization for soil fertility management and as a livestock feed source should be resolved by encouraging farmers to plant more forage trees on SWC structures, reducing the burden of crop residue as a livestock feed source.

**Author Contributions:** Conceptualization A.T., B.S. and M.B.; formal analysis, A.T.; investigation, A.T.; resources, A.T.; data curation, A.T.; writing—original draft preparation, A.T.; writing—review and editing, A.T., B.S. and M.B.; visualization, A.T.; supervision, B.S. and M.B.; project administration, B.S. and M.B.; funding acquisition, B.S. and M.B. All authors have read and agreed to the published version of the manuscript.

**Funding:** The authors would like to thank Addis Ababa University (AAU) for providing financial support for the data collection and write-up of the manuscript.

**Institutional Review Board Statement:** Not applicable.

**Informed Consent Statement:** Informed consent was obtained from all subjects involved in the study.

**Data Availability Statement:** Not applicable.

**Acknowledgments:** We would like to thank the farmers, agricultural development agents, and local administrators of the study area for their assistance during the field work. We are also grateful to Addis Ababa University (AAU) for providing the required facilities for the data analysis and write-up of this paper.

**Conflicts of Interest:** The authors declare no conflict of interest.

## Appendix A

**Table A1.** Comparison among CSA innovations of major components of livelihood vulnerability.

| CSA Innovations | Improved Variety | | | Crop Residue Management | | | Crop Rotation | | | Compost | | |
|---|---|---|---|---|---|---|---|---|---|---|---|---|
| | Adopter | Non-Adopter | *t* Value | Adopter | Non-Adopter | *t* Value | Adopter | Non-Adopter | *t* Value | Adopter | Non-Adopter | *t* Value |
| Natural disaster | 0.52 | 0.61 | −3.7 *** | 0.57 | 0.59 | −1.2 | 0.62 | 0.56 | 3.1 *** | 0.57 | 0.60 | −1.2 |
| Climate change | 0.65 | 0.72 | −3.7 *** | 0.73 | 0.67 | 3.2 *** | 0.75 | 0.67 | 3.9 *** | 0.69 | 0.71 | −0.7 |
| Exposure index | 0.58 | 0.67 | −4.54 *** | 0.64 | 0.63 | −0.76 | 0.68 | 0.61 | 4.18 *** | 0.63 | 0.65 | −1.2 |
| Ecosystem | 0.34 | 0.31 | 3.3 *** | 0.38 | 0.40 | −2.3 ** | 0.32 | 0.32 | 0.7 | 0.33 | 0.29 | 4.4 *** |
| Agriculture | 0.3 | 0.35 | −4.6 *** | 0.33 | 0.34 | −0.5 | 0.33 | 0.34 | −1.4 | 0.31 | 0.37 | −6.9 *** |
| Sensitivity index | 0.38 | 0.37 | 1.66 | 0.36 | 0.38 | −2.53 *** | 0.37 | 0.37 | 0.2 | 0.38 | 0.36 | 2.96 *** |
| Wealth | 0.71 | 0.62 | 6.9 *** | 0.65 | 0.71 | −5.25 *** | 0.64 | 0.71 | −4.7 *** | 0.65 | 0.74 | −6.7 *** |
| Technology/innovation | 0.90 | 0.76 | 19.2 *** | 0.81 | 0.80 | 0.5 | 0.81 | 0.80 | 0.7 | 0.82 | 0.79 | 3.0 *** |
| Infrastructure | 0.16 | 0.15 | 0.4 | 0.15 | 0.16 | −0.35 | 0.16 | 0.15 | 1.1 | 0.16 | 0.14 | −2.1 ** |
| Knowledge | 0.75 | 0.79 | −2.7 ** | 0.77 | 0.78 | −0.1 | 0.75 | 0.79 | −2.2 ** | 0.77 | 0.79 | −1.8 |
| Social network | 0.74 | 0.75 | −1.7 | 0.73 | 0.76 | −2.7 ** | 0.75 | 0.75 | 0.6 | 0.74 | 0.76 | −1.4 |
| Adaptive capacity index | 0.36 | 0.33 | 5.66 *** | 0.35 | 0.33 | 3.88 *** | 0.35 | 0.33 | 3.44 *** | 0.34 | 0.32 | 3.9 *** |
| IPCC-LVI = (E − AC) * S/100 | 0.09 | 0.12 | −6.96 *** | 0.106 | 0.112 | −0.8 | 0.12 | 0.10 | 3.16 *** | 0.106 | 0.115 | −1.33 |

| CSA Innovations | Row Planting | | | SWC | | | Agroforestry | | |
|---|---|---|---|---|---|---|---|---|---|
| | Adopter | Non-Adopter | *t* Value | Adopter | Non-Adopter | *t* Value | Adopter | Non-Adopter | *t* Value |
| Natural disaster | 0.57 | 0.62 | −1.9 * | 0.63 | 0.53 | 4.4 *** | 0.63 | 0.57 | 2.4 ** |
| Climate change | 0.68 | 0.76 | −3.5 *** | 0.73 | 0.66 | 3.55 *** | 0.73 | 0.69 | 1.6 |
| Exposure index | 0.62 | 0.68 | 3.1 *** | 0.67 | 0.59 | 4.98 *** | 0.67 | 0.62 | 2.45 ** |
| Ecosystem | 0.33 | 0.27 | 7.9 *** | 0.32 | 0.31 | 1.7 * | 0.35 | 0.31 | 4.8 *** |
| Agriculture | 0.32 | 0.39 | −7.9 *** | 0.33 | 0.34 | 1.2 | 0.30 | 0.34 | −4.5 *** |
| Sensitivity index | 0.38 | 0.34 | 4.61 *** | 0.37 | 0.37 | 0.2 | 0.38 | 0.37 | 2.39 ** |
| Wealth | 0.67 | 0.72 | −3.2 *** | 0.67 | 0.69 | 1.15 | 0.61 | 0.70 | −5.7 *** |
| Technology/innovation | 0.82 | 0.76 | 5.1 *** | 0.79 | 0.82 | −3.4 *** | 0.83 | 0.80 | 2.6 ** |
| Infrastructure | 0.16 | 0.14 | 1.5 | 0.16 | 0.15 | −1.0 | 0.16 | 0.15 | 1.3 |
| Knowledge | 0.76 | 0.81 | −3.1 *** | 0.76 | 0.79 | −1.9 * | 0.75 | 0.78 | −1.9 * |
| Social network | 0.74 | 0.76 | −1.6 | 0.74 | 0.75 | −1.1 | 0.73 | 0.75 | −1.7 |
| Adaptive capacity index | 0.34 | 0.32 | 2.87 *** | 0.34 | 0.33 | 1.8 * | 0.35 | 0.33 | 3.6 *** |
| IPCC-LVI = (E − AC) * S/100 | 0.106 | 0.123 | −3.47 *** | 0.123 | 0.097 | 2.2 ** | 0.123 | 0.107 | 0.16 |

Significance level * $p < 0.1$, ** $p < 0.05$, *** $p < 0.01$.

**Table A2.** Indicators of farm-level GHG emissions and adoption of CSA innovations.

| Indicators | Improved Variety | Crop Residue Management | Crop Rotation | Compost | Row Planting | SWC | Agroforestry |
|---|---|---|---|---|---|---|---|
| Wood lot (in ha) | 15 | 15 | 13 | 29 | 33 | 19 | 12 |
| Area of teff (in ha) | 154 | 243 | 197 | 323 | 361 | 253 | 107 |
| Teff yield (in qt/ha) | 21.1 | 27.8 | 20.7 | 42.0 | 47.8 | 26.8 | 13.6 |
| Area of maize (in ha) | 130 | 209 | 156 | 276 | 296 | 219 | 97 |
| Maize yield (in qt/ha) | 18.2 | 25.1 | 19.9 | 38.0 | 44.5 | 24.7 | 14.6 |
| Area of wheat (in ha) | 158 | 239 | 193 | 317 | 333 | 232 | 112 |
| Wheat yield (in qt/ha) | 22.2 | 27.3 | 22.1 | 41.5 | 47.7 | 26.2 | 16.5 |
| Area of potato (in ha) | 59 | 70 | 74 | 111 | 122 | 87 | 37 |
| Potato yield (in qt/ha) | 8.5 | 8.8 | 9.2 | 16.1 | 18.8 | 11.2 | 6.1 |
| Area of faba bean (in ha) | 28 | 37 | 31 | 50 | 54 | 40 | 17 |
| Faba bean yield (in qt/ha) | 4.5 | 4.7 | 4.6 | 8.1 | 9.1 | 5.1 | 2.6 |
| Area of barley (in ha) | 24 | 38 | 31 | 55 | 62 | 44 | 22 |
| Barley yield (in qt/ha) | 2.7 | 4.2 | 3.8 | 7.2 | 9.3 | 4.6 | 2.9 |
| Dairy cattle (in number) | 235 | 339 | 289 | 493 | 566 | 402 | 159 |
| Other cattle (in number) | 251 | 388 | 319 | 566 | 645 | 438 | 158 |
| Sheep (in number) | 304 | 480 | 398 | 760 | 883 | 607 | 203 |
| Goat (in number) | 60 | 63 | 50 | 94 | 109 | 60 | 37 |
| Horse (in number) | 44 | 86 | 60 | 113 | 129 | 101 | 28 |
| Poultry (in number) | 249 | 303 | 269 | 504 | 650 | 399 | 145 |
| Milk (in tons) | 7 | 15 | 9 | 20 | 20 | 15 | 6 |
| Meat (in tons) | 15 | 23 | 18 | 33 | 39 | 25 | 9 |
| Lime (in ton) | 15 | 22 | 19 | 34 | 34 | 192 | 10 |
| UREA (in tons) | 23.2 | 32.2 | 23.8 | 40.1 | 44.5 | 25.4 | 16.2 |
| NPS (in tons) | 51 | 61.2 | 44.5 | 93.8 | 103.8 | 48 | 45.1 |
| Firewood (in ton) | 489.9 | 716.0 | 584.7 | 1062.4 | 1192.5 | 803.4 | 363.5 |

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
