# Peer review of "Effectiveness of Climate-Smart Agriculture Innovations in Smallholder Agriculture System in Ethiopia"

_sustainability, doi:10.3390/su142316143_

Round 1

Reviewer 1 Report

This manuscript reports on soil erosion control. However, the novelty or uniqueness of the research is unclear.  How is this research different from previously published research? In addition, your submitted manuscript should be carefully reviewed for typos and logical flow.

- In the abstract, the take-home message is missing. Please revise the sentences to clarify the research question.

-- Authors are advised to enhance the discussion.

- Please re-edit your manuscript. Most of the sentences are not understandable. 

Author Response

Dear Reviewer

First of all, this research is not about soil erosion.  may be the manuscript you recieved is not "Effectiveness of climate smart agriculture innovations in smallholder agriculture system" so please request the right manuscript from the editor.

Reviewer 2 Report

The present article “Effectiveness of climate smart agriculture innovations in small holder agriculture system” deals an interesting current topic of  climate change , Authors have comprehensively compiled the topic. However before acceptance I have  some quarry and suggestions, which must be resolved.

-Author should avoid the large sentences and  also correct the  english throughout the article  

-The reference  style of the manuscript is not of MDPI style, Use number system in the text

-Line 109-   located between 9o 38' 00" and 10o55' 24" North latitude and 37o 07' 00" to 38o 17' 00". Please update the unit of degree (°) not 0 through out the MS.

-Line no-124-126 : Quantitative data was gathered using a structured household survey questionnaire 124 that focused on household and farm characteristics was collected on a one-to-one inter- 125 view basis by well-trained and experienced enumerators using android tablets- reframe the sentence, grammatically incorrect.

-Line 160-170, check and reframe the sentences for clarity

-Line177-182-  formula or text mentioned in not understandable.

Author Response

Dear Reviewer

I will revised the manuscript as you suggested. Thank you

Reviewer 3 Report

The manuscript “Effectiveness of climate-smart agriculture innovations in smallholder agriculture system” deal with the effectiveness of climate-smart agriculture innovations in smallholder agriculture system in Ethiopia, and will be interesting for Sustainability readers, after revision.

The scientific problem is interesting, using experienced enumerators using android tablets for agricultural research, but what is the particular sustainable aspect for agriculture in this study, please specify in the aim/discussion/conclusion.

Methods: Please describe point by point the econometric model used in this research;

It is difficult to understand what is the main objective of the research, what is the specific conclusion, and what can be cited.

The authors should be more connected to the aims and the conclusions of the manuscript.

The conclusion must be improved, and more detailed, perhaps on a few points (Carbon balance, and another).

The multiplicity of abbreviations, including in the summary, makes it difficult to read the text and possibly cite it by other researchers.

The authors should try harder to exploit/present the great potential of the research.

Some specific comments:

23: The econometric results… - The econometric analysis/method results will be better;

25: adoption – as a keyword is unclear, should be specified, why adoption;

138: 2.3. Measurements – Carbon balance?             

177-182: (1) (2) (3) - will be improve;

183: 2.4. Data analysis – will be described in more detail;

685, and below: Table 1. Continued. – without, bracelets in the table, spaces – improve the intervals;

600: This study aims to contribute to the literature from the… - remove;

Reviewer 4 Report

The manuscript entitled “Effectiveness of climate smart agriculture innovations in small holder agriculture system” authored by Teklu et al. intends to evaluate climate smart agriculture (CSA) innovations impact on enhancing food security, reducing vulnerability, and reducing farm level GHG emissions. The MS elaborates an undefined survey methodology to draw conclusions on the subject matter. The abstract seems to be of report writing rather an original research article. Results have been stated as generalized statements with no concrete data based findings. The most serious concern is regarding the methodology as authors have employed an ambiguous regression model to analyze the survey based findings. The study seems to be highly localized with meager global pertinence in terms of methodology and research findings. Recommendation in abstract has been illustrated as a generalized statement with robust support of research data. Authors have not sufficiently established study rationale rather introduction section gives the concurrence of random statements. Additionally, I could not locate the study hypothesis which needs to be described just before stating the objectives of the study. Authors are supposed to objectively analyze the peer-findings relevant to protocols/methodologies adopted during the course of study but this aspect is seriously lacking, e.g., food consumption score etc. Data analysis skips critical information pertaining to software used and test employed for data analysis. Results and discussion section entails very surficial results description while most of discussion focuses on irrelevant details. Figures lack statistical symbols for showing any sort of significance among recorded data. Conclusion encompasses too many generalized statements and direly needs to be imparted the essence of briefness and comprehensiveness. As per my observation, the MS in current form does not meet the requisites of the journal.    

Round 2

Reviewer 2 Report

Author have  extensively  revised the article. I recommend acceptance in the current form.  

Author Response

I have made enough revisions

Reviewer 3 Report

Please respond point by point to the reviewer's comments. There is no cover letter.

Author Response

We have corrected all comments according to your request. Would you please use the button on the menu bar under Review and Open the Accep button under the Title bar to open  " Accept all changes in the document" and see the latest revison of our manuscript. Thank you.

Reviewer 4 Report

The manuscript authored by Teklu et al. entitled “Effectiveness of climate smart agriculture innovations in small holder agriculture system” elaborates an undefined survey methodology to draw conclusions on the subject matter. I had previously reviewed the MS and suggested few critical suggestions. Authors have improved the MS but still following points needs further consideration.

My prime concern remains about abstract that seems to be of report writing rather an original research article. Results have been stated as generalized statements with no concrete data based findings. The most serious concern is regarding the methodology as authors have employed an ambiguous regression model to analyze the survey based findings. The study seems to be highly localized with meager global pertinence in terms of methodology and research findings. Recommendation in abstract has been illustrated as a generalized statement with robust support of research data. Authors have not sufficiently established study rationale rather introduction section gives the concurrence of random statements. Additionally, I could not locate the study hypothesis which needs to be described just before stating the objectives of the study. Authors are supposed to objectively analyze the peer-findings relevant to protocols/methodologies adopted during the course of study but this aspect is seriously lacking, e.g., food consumption score etc. Data analysis skips critical information pertaining to software used and test employed for data analysis. Results and discussion section entails very surficial results description while most of discussion focuses on irrelevant details. Figures lack statistical symbols for showing any sort of significance among recorded data. Conclusion encompasses too many generalized statements and direly needs to be imparted the essence of briefness and comprehensiveness.

Overall, authors have skipped the essence of a scientific article which is briefness and comprehensiveness, while article has been written in dissertation and report writing style. I shall sustain my observation of non-acceptability of this MS in the revised form.

Author Response

I have made enough revisions